# Winter Wheat Nitrogen Estimation Based on Ground-Level and UAV-Mounted Sensors

**DOI:** 10.3390/s22020549

**Published:** 2022-01-11

**Authors:** Xiaoyu Song, Guijun Yang, Xingang Xu, Dongyan Zhang, Chenghai Yang, Haikuan Feng

**Affiliations:** 1Information Technology Research Center, Beijing Academy of Agriculture and Forestry Sciences, Beijing 100097, China; Songxy@nercita.org.cn (X.S.); Yanggj@nercita.org.cn (G.Y.); Xuxg@nercita.org.cn (X.X.); 2Key Laboratory of Quantitative Remote Sensing in Agriculture of Ministry of Agriculture, Beijing 100097, China; 3Anhui Engineering Laboratory of Agro-Ecological Big Data, Anhui University, Hefei 230601, China; 4Aerial Application Technology Research Unit, USDA-Agricultural Research Service, College Station, TX 77845, USA; chenghai.yang@usda.gov; 5National Engineering Research Center for Information Technology in Agriculture, Beijing 100097, China; Fenghk@nercita.org.cn

**Keywords:** leaf nitrogen concentration, plant nitrogen content, nitrogen nutrition index, Gaussian process regression

## Abstract

A better understanding of wheat nitrogen status is important for improving N fertilizer management in precision farming. In this study, four different sensors were evaluated for their ability to estimate winter wheat nitrogen. A Gaussian process regression (GPR) method with the sequential backward feature removal (SBBR) routine was used to identify the best combinations of vegetation indices (VIs) sensitive to wheat N indicators for different sensors. Wheat leaf N concentration (LNC), plant N concentration (PNC), and the nutrition index (NNI) were estimated by the VIs through parametric regression (PR), multivariable linear regression (MLR), and Gaussian process regression (GPR). The study results reveal that the optical fluorescence sensor provides more accurate estimates of winter wheat N status at a low-canopy coverage condition. The Dualex Nitrogen Balance Index (NBI) is the best leaf-level indicator for wheat LNC, PNC and NNI at the early wheat growth stage. At the early growth stage, Multiplex indices are the best canopy-level indicators for LNC, PNC, and NNI. At the late growth stage, ASD VIs provide accurate estimates for wheat N indicators. This study also reveals that the GPR with SBBR analysis method provides more accurate estimates of winter wheat LNC, PNC, and NNI, with the best VI combinations for these sensors across the different winter wheat growth stages, compared with the MLR and PR methods.

## 1. Introduction

Nitrogen (N) is a crucial nutrient required for crop growth and grain formation. Agricultural managers can regulate N management at suitable rates and opportune moments based on the crop’s N requirements. As an essential scatheless and real-time technique, remote sensing technologies have been proved valuable for crop N status evaluation [1,2,3,4,5,6].

Much meaningful progress in sensor technology for evaluation of plant N status has been achieved in recent years. Leaf sensors, such as chlorophyll meters [7,8,9] and Dualex sensors [10], have been widely used to measure crop N status. These leaf clip sensors show a stable relation with plant N due to their direct contact with the plant. However, there are also some restrictions on these sensors because plant leaf water content, leaf structure, or other nutrient deficiencies may also easily influence the sensor readings [11]. Some studies indicate that measurement of polyphenol concentrations in the leaf is a new method to overcome such barriers [12]. The ratio of chlorophyll to polyphenol is more stable than the leaf chlorophyll distribution [11,13]. At the canopy level, the spectral features of chlorophyll from the visible to near-infrared bands are used as indicators of crop N [14,15,16]. The Multiplex 3 portable sensor has also been widely used for plant N diagnosis in recent years, and it can detect plant chlorophyll and flavonol compounds simultaneously through the chlorophyll fluorescence method [17]. As a new exploration of the low-altitude remote sensing method, unmanned aerial vehicle (UAV)-based sensing has been widely used in different fields recently due to its flexibility, affordability, and applicability for large-scale monitoring compared to handheld active sensing [18,19]. Digital color images acquired by UAV or airplane-mounted sensors have also proven to be a feasible way to estimate canopy variables such as leaf chlorophyll content [20], nitrogen status [21], wheat senescence [22], and vegetation cover [23] in a large area [24]. In general, rapid, precise, and non-destructive acquisition of N information has become an essential technique for crop nutrition and growth diagnosis [5,25], which helps dynamic regulations of N fertilizer use [26].

Previous studies attempted to retrieve N via a radiative transfer model. As an example, Jacquemoud et al. attempted to incorporate N into the PROSPECT model, but this approach was abandoned because of inconsistencies in N retrieval through model inversion [27]. More efforts have been directed at estimating crop N status using empirical methods based on observed reflectance to measured vegetation characteristics [28,29]. Many practical regression techniques using hyperspectral bands, vegetation indices (VIs), and different types of sensor data have been proved influential in plant N estimation [21,28,29]. Parametric regression (PR) methods based on band information or VIs of broadband satellite sensors are probably the oldest and largest group of variable estimation approaches [30]. VIs enhance spectral features sensitive to a vegetation property while reducing disturbances by combining some spectral bands [31,32]. The major advantage of VIs is their intrinsic simplicity. However, the selection of an optimal subset of hyperspectral bands or best VIs sensitive to plant N encounters both numerical and computational difficulties [33,34]. Multivariable linear regression (MLR) methods are attractive because of their fast performance in coping with spectroscopic data and their typical reliance on the estimation of covariances [30]. Miphokasap et al. demonstrated that the model developed by MLR led to a higher correlation coefficient and lower errors than model applications based on narrowband VIs in estimating canopy nitrogen [30,35]. In recent decades, a variety of non-linear, non-parametric methods have been developed, going beyond linear regression or linear transformation techniques. These methods, also referred to as machine learning regression algorithms, apply non-linear transformations [30]. Gaussian process regression (GPR) applied to spectroscopic and hyperspectral data started more recently, such as airborne HyMap mapping of leaf chlorophyll content [36] and spaceborne CHRIS mapping of leaf chlorophyll content, LAI, and fractional vegetation content [37]. The GPR method is deemed to be one of the most exciting machine learning regression algorithms, which can provide a full conditional statistic for the predicted variable [38,39]. GPR with the sequential backward feature removal (SBBR) routine proposed by Verrelst et al. leads to the identification of the most sensitive vegetation index for the crop nitrogen variable [34].

Nevertheless, when this technique is applied in studies with spectral data from different sensors, results on the best VIs or combinations and their performance on crop nitrogen diagnosis have rarely been reported. Therefore, in this study, we propose assessing the performance of different sensors for winter wheat N status estimation through PR, MLR, and GPR analyses. To determine which spectral features could sufficiently estimate crop N status indicators, we considered one leaf N status indicator (i.e., the leaf N concentration (LNC)), one plant N status indicator (i.e., the N plant concentration (PNC)), and one relative plant N indicator (i.e., the nutrition index (NNI)), which is calculated as the ratio of the measured N concentration and the critical N concentration. In this study, we also performed sensitivity analysis on the wheat N indicators for different sensors through the PR, MLR, and GPR methods. The main objectives of this study were to: (1) evaluate the ability of four different proximal and UAV-mounted sensors for winter wheat N status estimation; (2) identify the optimal VIs and VI combinations derived from the sensors for accurate winter wheat N status estimation; and (3) determine whether GPR models based on the combination of VIs could further improve the estimation accuracy.

## 2. Materials and Methods

### 2.1. Experimental Design

This study was carried out at the National Experimental Station for Precision Agriculture in the Changping District of Beijing, China (40°10.6′ N, 116°26.3′ E; Figure 1). The area of the study field was about 0.80 ha. Winter wheat was sown on 4 October 2013, in 15 cm-wide rows. The variety of wheat was Jindong 22. A total of 8 different nitrogen fertilizer treatments were assigned to 104 plots in this experiment, each plot measuring 7 × 7.5 m (Figure 1). There was a 1 m buffer around each fertilization plot to minimize the interference between different fertilizer treatments. Base fertilizer, including 72 kg/ha N, 60 kg/ha P_2_O_5_, and 75 kg/ha K_2_O, was applied to the experiment plots when the wheat was seeded, except for the CK treatments. Topdressing was applied at the wheat jointing stage (16 April, corresponding to Feekes 6), and detailed fertilizer information about the treatments is listed in Table 1.

### 2.2. Sensor Data Collection

The leaf and canopy spectral parameters were measured in a 1 m^2^ rectangular region within each plot at the wheat raising stage (8 April 2014, corresponding to Feekes 5) and the filling stage (28 May 2014, corresponding to Feekes 11). In this study, we used four different optical instruments to collect wheat spectral data. The Dualex sensor was used to measure wheat leaf spectral parameters. The Multiplex sensor and the ASD field spectrometer were used to collect wheat canopy spectral parameters. A Sony DSC-QX100 digital RGB camera mounted on a DJI S1000 UAV was used to collect wheat images for the experiment fields. Table 2 lists the detailed information on the sensors used in this study.

#### 2.2.1. Leaf- and Canopy-Level Data Collection

For each plot, 10 wheat plants were randomly selected for Dualex measurement. The first and second fully expanded leaves of the plants were clipped at the middle location, and the measurement averages represented the Duelax NBI, Chl, and FLAV values of the plot. Multiplex measurements and ASD spectral measurements were performed and repeated 20 times at 10 cm and 1.3 m above the wheat canopy, respectively. The measurement averages represented the wheat canopy fluorescence and reflectance values for each plot. A total of 9 Multiplex fluorescence VIs and 28 ASD spectral VIs sensitive to plant N are listed in Table 3.

#### 2.2.2. UAV-Level Data Collection

The UAV digital images were obtained at a 50 m altitude. Pix4Dmapper software (Pix4D Inc., Lausanne, Switzerland) was used for automatic image mosaicking. The original digital number (DN) values of the images were calibrated by the experimental line method (Equation (1)) [63]:(1)$${\mathrm{DN}}_{i}=\frac{{\mathrm{DN}}_{\mathrm{object}}-{\mathrm{DN}}_{\mathrm{black}}}{{\mathrm{DN}}_{\mathrm{white}}-{\mathrm{DN}}_{\mathrm{black}}}\times 255$$
where DN_i_ represents the calibrated DN value for band i, such as R, G, and B; DN_object_ is the original DN value of the images; and DN_white_ and DN_black_ are the original DN values from the white and black panels in the UAV-DN images. Then, the images were resampled to 2 cm using the nearest neighbor method. An area of interest (AOI) (2 × 2 m) located in the middle region of each sample plot was delineated on the image to exclude the border effect. A total of 208 winter wheat AOIs (104 AOIs at wheat Feekes stage 5 and 104 AOIs at wheat Feekes stage 11) were clipped for the subsequent analysis. Then, the spectral data from the mix of winter wheat and soil background were extracted, and the VIs were calculated to indicate the winter wheat growth status for each plot. Table 4 lists the RGB image bands and VIs used in this study.

### 2.3. Plant Sampling Data Collection

We collected the plant samples as soon as the sensors’ measurements were completed. In this study, plants from an area of 0.12 m^2^ (two rows and 40 cm wide) near the spectral measurement plot were collected and sent to the laboratory for analysis. The vegetative organs (leaves, stems, and ears) were then divided and dried in the oven at 80 °C for 24 h. The biomass (g/m^2^) for wheat leaves (leaf biomass, LB), stems (stem biomass, SB), and ears (ear biomass, EB) per unit area was calculated based on the measured planting density and the dry weight of the samples. The leaf nitrogen concentration (LNC % (g N 100 g^−1^ DM)), stem nitrogen concentration (SNC % (g N 100 g^−1^ DM)), and ear nitrogen concentration (ENC % (g N 100 g^−1^ DM)) were then determined using the standard Kjeldahl method [67]. Plant N accumulation (PNA, kg/ha) was the sum of leaf, stem, and ear N accumulation (Equation (2)). Wheat plant N content (PNC %) was determined by PNA (kg/ha) divided by the plant biomass (kg/ha) (Equation (3)) [68].
(2)PNA (kg/ha)=(LB×LNC) +(SB×SNC)+(EB×ENC)
(3)PNC (%)=PNA/(LB+SB+EB)

The NNI was used to indicate the N status for each experimental plot, independent of the growth stage and differing biomass levels. The NNI was calculated by Equation (4) [69]:(4)$$\mathrm{NNI}\;=N_{act}/N_{c}\;$$
where $N_{act}$ is the actual N concentration, and *N_c_* is the critical N concentration as a percentage of the aboveground dry matter expressed in % DM. The $N_{c}$ for winter wheat was described by Equation (5) [70]:(5)$$N_{c}=4.15\times W^{-0.38}$$
where *W* is the aboveground biomass, expressed in Mg dry matter (DM) ha^−1^.

### 2.4. Data Analysis Methods

Five traditional PR methods (linear, logarithmic, power, exponential, and second-order polynomial) and the MLR and GPR methods were used to establish relationships between wheat N parameters and spectral VIs in this study. For the PR methods, 70% of the samples were used to build the models, and 30% were used to validate them. A stepwise MLR was used to estimate the winter wheat N status by recursively applying multiple regression.

Compared to other machine learning regression approaches, the parameter optimization of GPR is simpler and can be automatically completed by maximizing the marginal likelihood in the training set [34]. The core of the kernel method for GPR is the squared exponential. The use of a flexible kernel function (covariance function) generally suffices for tackling most regression problems and is beneficial if prior knowledge is weak [37]. This is an opportunity to exploit asymmetries in the feature space by including a parameter per feature, as in the very common anisotropic squared exponential (SE) kernel function:(6)$$K\left(x_{i},x_{j}\right)=\gamma exp(-\overset{B}{\underset{b=1}{\sum }}\frac{{\left(x_{i}^{b}-x_{j}^{b}\right)}^{2}}{2\sigma_{b}^{2}})$$where γ is a scaling factor; $\sigma_{b\;}$ is the length scale per input feature, *b* = 1, …, *B*; and $x_{i}^{b}$ represents the *b*th feature of the input feature vector $x_{i}$. A smaller value of $\sigma_{b\;}$ indicates a higher level of informative content of this certain image feature. Verrelst et al. [34] integrated this property and SBBR as a new feature selection algorithm, in which the least significant feature with the highest $\sigma_{b\;}$ was removed at each iteration and a new GPR model was retrained with the remaining features only. A feature combination would eventually be identified by the feature selection algorithm. To enable the automated identification of the best-performing features for any regression problem, this feature selection algorithm was automated and integrated into a user-friendly tool named GPR-BAT [34].

In this study, GPR with SBBR was used firstly to identify the best VI combinations for different N parameters, and then MLR and GPR were used to build N parameter estimation models based on those VI combinations. The 10-fold cross-validation technique was used to determine the optimal number of latent factors based on the lowest root mean square error (RMSE). The accuracy of each model was evaluated by the coefficient of determination R^2^, RMSE, the mean absolute error (MAE), and the Nash–Sutcliffe modeling efficiency (NSE). The equations used to calculate RMSE, MAE, and NSE are presented in Table 5.

## 3. Results

### 3.1. Variation in Winter Wheat N Indicators

Table 6 describes the variation in the winter wheat N indicators for different growth stages. PNC had the most variation (CV = 49.46% across growth stages), followed by LNC (CV = 32.34%) and NNI (CV = 18.59%). At Feekes stage 5, the greatest difference occurred in NNI, which ranged from 0.74 to 1.58. PNC had more variation at Feekes stage 5 (CV = 13.42%) than at Feekes stages 11 (CV = 8.99%). The CV for LNC increased from 11.30% to 13.42% from Feekes 5 to Feekes 11. The CV for NNI decreased from 17.17% at Feekes 5 to 11.97% at Feekes 11.

### 3.2. Relationships between Wheat N and VIs for Different Sensors

After the correlation analysis between the VIs and the three N parameters, the VI with the highest correlation coefficient was selected to establish the N estimation model. Table 7 lists the VIs and their R^2^ and RMSE for the best models for wheat LNC, PNC, and NNI through traditional PR methods. For the PR methods, 70% of the samples (70 samples) were used to build the models, and 30% (34 samples) were used to validate them. It could be observed, at wheat Feekes stage 5, that the Dualex- and Multiplex-based VIs had a stronger correlation with wheat LNC, PNC, and NNI. The R^2^ for the LNC and PNC models based on the Dualex NBI reached 0.80 and 0.79, respectively. At the late wheat growth stage, canopy-level VIs performed better than the leaf-level VIs. The RGB and ASD VIs performed better than the Dualex and Multiplex VIs for LNC, PNC, and NNI estimation. RGB camera-based r performed best for LNC estimation at wheat Feekes stage 11, followed by ASD mND_705_, and Dualex CHI. For PNC estimation, ASD NDRE performed best, followed by the RGB camera-based ExR. Multiplex performed best on the wheat N status estimation across wheat stages (Feekes 5–11), and the R^2^ reached 0.87 for BRR_FRF and LNC, 0.86 for BRR_FRF and PNC, and 0.56 for NBI_G and NNI.

### 3.3. Wheat N Estimation through UAV-Level VIs

GPR and MLR were used to estimate the relationships between the RGB camera VIs and N variables. The 10-fold cross-validation technique was used to determine the optimal number of latent factors based on the lowest root mean square error (RMSE). Table 8 lists the cross-validation results for models built by the selected VI combinations and all VIs through the GPR and MLR methods. The VI combinations for different N variables were selected by the GPR-SBBR feature selection algorithm.

It can be observed in Table 8 that the best VI combinations for LNC, PNC, and NNI estimation at the different wheat growth stages are different. At Feekes stags 5, the VI combinations selected by the GPR-SBBR algorithm for LNC, PNC, and NNI estimation were all B and b, indicating that the RGB camera VIs B and b can be used to identify the N status at the wheat growth stage.

GPR models built by the selected VIs had a higher cross-validation R^2^ compared to the GPR models built by all VIs. MLR models based on the selected VIs achieved a higher R^2^ at Feekes stage 5 and Feekes stage 11 for LNC and NNI estimation. However, MLR models built by all VIs performed better than MLR models built by the selected VIs on N status estimation across different wheat growth stages (Feekes stage 5–11).

### 3.4. Wheat N Estimation through Canopy-Level VIs

Table 9 and Table 10 list the cross-validation results for the N estimation models through GPR and MLR based on canopy-level VIs. Table 9 indicates that for LNC estimation, the best ASD VI combinations were TBI1, NDIopt, and TBI2 at wheat Feekes stage 5, SR_(700,670)_, SR_(418,405)_, and SR_(740,720)_ at Feekes stage 11, and SR_(418,405)_, TBI1, and PPR across Feekes 5–11. For PNC estimation, the optimal VI combinations were TBI1, NDIopt, and TBI2 at wheat Feekes stage 5, SR_(700,670)_, SR_(418,405)_, and SR_(740,720)_ at Feekes stage 11, and TBI1, PPR, REV, and REFD across Feekes 5–11. The best VIs for NNI estimation included two VIs (MSAVI and PPR) at wheat Feekes stage 5, four VIs (NDWI, MCARI, MSAVI, and PVR) at wheat Feekes stage 11, and five VIs (SR_(418,405)_, NDIopt, MSAVI, MCARI1, and PPR) at wheat Feekes stage 5–11. It can be seen in Table 9 that the LNC, PNC, and NNI estimation models built by GPR with the selected VIs performed better than those built by the other three methods. MLR with all ASD VIs had a higher cross-validation R^2^ than MLR with the selected ASD VIs or GPR with all ASD VIs at wheat Feekes stage 5 and Feekes stage 11.

Table 10 lists the LNC, PNC, and NNI estimation results through both the GPR method and the MLR method. The best-performing Multiplex VI combinations for LNC estimation were SFR_G, FLAV, and NBI_R at wheat Feekes stage 5, NBI_R and SFR_R at wheat Feekes stage 11, and SFR_R, BRR_FRF, and NBI_R at wheat Feekes stage 5–11. The best Multiplex VI combinations for PNC were SFR_R, FLAV, and NBI_R at wheat Feekes stage 5, SFR_R at wheat Feekes stage 11, and SFR_R, BRR_FRF, and NBI_G at wheat Feekes stage 5–11. For NNI, Multiplex FLAV and FER_RG at wheat Feekes growth stage 5, SFR_G and SFR_R at wheat Feekes growth stage 11, and SFR_G, SFR_R, and NBI_G at wheat Feekes stage 5–11 performed well. The cross-validation R^2^ values for LNC, PNC, and NNI estimated by GPR with the selected VIs were higher than those estimated by GPR or MLR with all VIs at wheat Feekes stage 5 and Feekes stage 11.

### 3.5. Wheat N Estimation through Leaf-Level VIs

Table 11 lists the cross-validation results for the three N parameters estimated through the GPR and MLR methods based on Dualex VIs. It can be seen in Table 11 that the best-performing VI for LNC, PNC, and NNI was NBI at wheat Feekes stage 5. At wheat Feekes stage 11, Chl performed best for LNC, PNC, and NNI estimation. These results indicate that the Dualex sensor can detect the leaf N status at different wheat growth stages, especially at Feekes stage 5. During Feekes stage 5–11, the combination of the Dualex NBI and Chl performed best for estimating LNC, PNC, and NNI.

## 4. Discussion

Ground-based spectrometers are considered capable of monitoring crop trait expressions and nutrients [74,75]; however, data collection with handheld sensors has the limitations of instability, low sensing efficiency, and high cost [76]. In this study, variable-rate N fertilization treatments in winter wheat were carried out to represent the reality, which resulted in different winter wheat N nutrition conditions (Table 1 and Table 6). Four different sensors were used to obtain crop growth information during two winter wheat critical growth stages. This allowed a preliminary assessment of the utility of the handheld leaf and canopy sensors as well as the UAV-mounted sensor in the estimation of N status indictors of winter wheat. The predictive performances of the four sensors were compared using different modeling methods (PR, MLR, and GPR).

### 4.1. N Estimation Comparison for Different Sensors

We investigated the wheat N estimation capability for different sensors by comparing the cross-validation R^2^ of the models. Figure 2 clearly shows that different sensors have different capabilities in N status detection during the wheat growth stage.

It can be observed in Figure 2a,b that the Dualex sensor had an obvious advantage in LNC and PNC estimation at wheat Feekes stage 5, and the R^2^ for different modeling methods varied from 0.77 to 0.80 for LNC, and from 0.78 to 0.79 for PNC. The Multiplex sensor can also detect LNC and PNC better than the ASD spectrometer or the RGB camera at the early wheat growth stage because it can eliminate erroneous signals from bare soil or distinguish between different N treatments in shadow with full sunlight. At Feekes stage 11, due to the biomass saturation, the correlations between N status and the VIs of the Dualex and Multiplex sensors decreased quickly. The modeling mean R^2^ decreased from 0.79 to 0.33 for Dualex, and from 0.58 to 0.32 for Multiplex, over the winter wheat growth stages. Compared with the other sensors, the ASD spectrometer performed well in LNC and PNC estimation through GPR with the selected VIs at winter wheat Feekes stage 11. The estimation R^2^ reached 0.51 for LNC and 0.49 for PNC. For NNI detection, the Dualex sensor worked relatively well at wheat Feekes stage 5, with a mean R^2^ of 0.50 for different regression methods, followed by the Multiplex sensor, with a mean R^2^ of 0.48 for different regression methods. All sensors did not perform well in NNI estimation at wheat Feekes stage 11 in this study. When data from the two stages were combined (Feekes 5–11), the modeling accuracy for the three N parameters all increased obviously. It can be observed in Figure 2 that the ASD and Multiplex sensors can detect LNC and PNC better than the RGB camera and the Dualex sensor. The R^2^ values are all greater than 0.90 for the ASD and Multiplex sensors.

Numerous studies have revealed that, at the leaf level, there is a well-established correlation between photosynthetic capacity and N content [27], while at the scale of canopies, reflectance patterns represent the integrated effects of leaf water content, biochemical constituents, and various components of the plant structure [77]. As reviewed by Tremblay et al., determining N status in plants by means of chlorophyll fluorescence can overcome some of the limitations of reflectance-based chlorophyll methods [10]. The Dualex VIs, namely, the Nitrogen Balance Index (NBI) and the chlorophyll index (CHI), seem to be good indicators to evaluate the N conditions in wheat at the early growth stage. This was also confirmed by the results of this study. The Multiplex sensor can overcome the deficiency of a chlorophyll meter because it is able to distinguish N treatments equally well in shadow or full sunlight, and at any time during the day [10]. Multiplex performs better in estimating wheat LNC, PNC, and NNI than ASD at the early wheat growth stage, while ASD can detect the wheat N indicators more accurately than Multiplex at the late wheat growth stage.

### 4.2. Accuracy Evaluation of GPR, MLR, and PR Methods

In this study, the GPR model combined with the SBBR algorithm was used to obtain the optimum N-sensitive VI combinations for different sensors. Then, both the MLR and GPR models based on the VI combinations and all VIs (Table 8, Table 9, Table 10 and Table 11), as well as the PR model (Table 7), were investigated for N estimation accuracy through comparing the models’ R^2^ and RMSE. Figure 3 and Figure 4 show the mean absolute error (MAE) and the Nash–Sutcliffe modeling efficiency (NSE) for LNC, PNC, and NNI through the PR, MLR, and GPR modeling methods.

The ideal MAE value is 0 from the range of 0 to positive infinity, while the optimum NSE value is 1 from the range of negative infinity to 1. As shown in Figure 3, the MAE values for the LNC models built by GPR methods for different sensors are generally less than those of the PR and MLR methods. Meanwhile, the NSE values in Figure 4 indicate that the GPR methods have a higher accuracy, especially when modeling with the ASD and RGB VIs across different wheat growth stages. For PNC estimation, the PR method can obtain a lower MAE at wheat Feekes stage 5, while the GPR models perform better at wheat Feekes stage 11. For NNI estimation, GPR modeling with the selected Multiplex and ASD VIs has a higher MAE than the other methods, indicating that these two sensors may not be suitable for NNI estimation at the early wheat growth stage. It can be seen in Figure 3 and Figure 4 that when the two stages are combined, GPR modeling with all VIs or the selected VIs can estimate N parameters with a lower MAE and a higher NSE than the other modeling methods.

Figure 5 shows the scatterplots of the estimated vs. measured LNC, PNC, and NNI through the GPR-SBBR process based on the best VIs for the four sensors.

Contrary to other methods, the training phase in GPR takes place in a Bayesian framework. Verrelst et al. compared GPR with parametric methods based on established and generic VIs [34]. GPR not only outperformed parametric linear regression methods but also offered additional interesting features that go far beyond what is typically available from parametric or non-parametric approaches [35]. In this study, the GPR model with the selected ASD VIs yielded an obviously higher accuracy for LNC and PNC estimation compared to the PR and MLR models. The result from this study is consistent with previous findings.

Apart from the GPR model, the R^2^, MAE, and NSE value variation for the MLR and PR models across different wheat growth stages indicates that these models could not produce reliable estimates of winter wheat N status indicators. RGB image-based VIs were used to estimate the N status of winter wheat in field conditions. The observed wheat N parameters were fairly well described by the image data with the GPR model, indicating that a digital camera can be used as a low-cost tool to estimate crop N status and deliver information for N management over a broader region.

The relationships between the VIs from different sensors and N status indicators at the early wheat growth stage were generally better than those at the late stage. The growth stage was found to have a significant effect on the relationships between VIs and N status indicators. This was also confirmed by the results of previous studies [3,38]. However, because the Multiplex or ASD sensor requires measurements made over the plant, covering a small area, it is difficult to apply this type of sensor in monitoring N status in larger areas. Although multispectral or hyperspectral images with near-infrared sensitivity can often convey more information than conventional RGB images, RGB cameras have been more frequently used due to their low cost and ease of use [78]. In this study, RGB image-based VIs were used to estimate the nitrogen status of winter wheat canopies under field conditions. Compared with the other sensors, the UAV-mounted RGB camera is a better selection for crop growth and nutrition monitoring due to its ability to conduct the same type of assessment over hundreds or thousands of plots with a low cost and ease of use, as shown in this study. Different types of sensors can be carried by the UAV to detect the crop growth and nutrition status. The crop information extracted from UAV images is affected not only by the sensor type, but also by the UAV imaging conditions, such as the UAV flight height and speed. In this study, only one sensor (the RGB camera) was mounted on the UAV to test its ability to estimate crop N status. More research is needed to explore the feasibility of this type of camera and other sensors for estimating the spatial distribution of the N status of winter wheat and other crops under different UAV flight heights and speeds over large areas.

## 5. Conclusions

Crop nitrogen nutrition diagnosis is important for practicing precision farming. Proximal and airborne sensors provide useful information for the diagnosis of crop N nutritional status. In this study, a GPR method with the 10-fold cross-validation SBBR routine was used to identify the best VIs sensitive to wheat LNC, PNC, and NNI for four different sensors. The observed wheat N parameters were well described by the GPR and traditional PR methods throughout the two growth stages. The following conclusions can be drawn from this study:

At the early wheat growth stage, the Dualex NBI is a good leaf-level indicator for wheat LNC, PNC, and NNI estimation. Meanwhile, the combination of Multiplex SFR_G, FLAV, and NBI_R, the combination of SFR_R, FLAV, and NBI_R, and the combination of FLAV and FER_RG are the best canopy-level indicators for LNC, PNC, and NNI estimation at the early wheat growth stage. The results indicate that the optical fluorescence sensor provides more accurate estimates of winter wheat N status at a low-canopy coverage condition.

At the late wheat growth stage, the best ASD VIs, including the combination of SR_(700,670)_, SR_(418,405)_, and SR_(740,720)_, the combination of SR_(418,405)_, NDIopt, RDVI, and REFD, and the combination of NDWI, MCARI, MSAVI, and PVR, provided accurate estimates for wheat LNC, PNC, and NNI. The results indicate that the ASD sensor, which collects more detailed information, is an essential tool for N estimation at the late wheat growth stage, although the reflectance patterns of ASD represent the integrated effects of leaf water content, biochemical constituents, and various components of the plant structure.

This study reveals that the GPR with SBBR method provides more accurate estimates of winter wheat LNC, PNC, and NNI with the best VI combinations of Dualex, Multiplex, ASD, and RGB camera sensors, compared with GPR with full VIs or the MLR and PR methods.

## Figures and Tables

**Figure 1 sensors-22-00549-f001:**
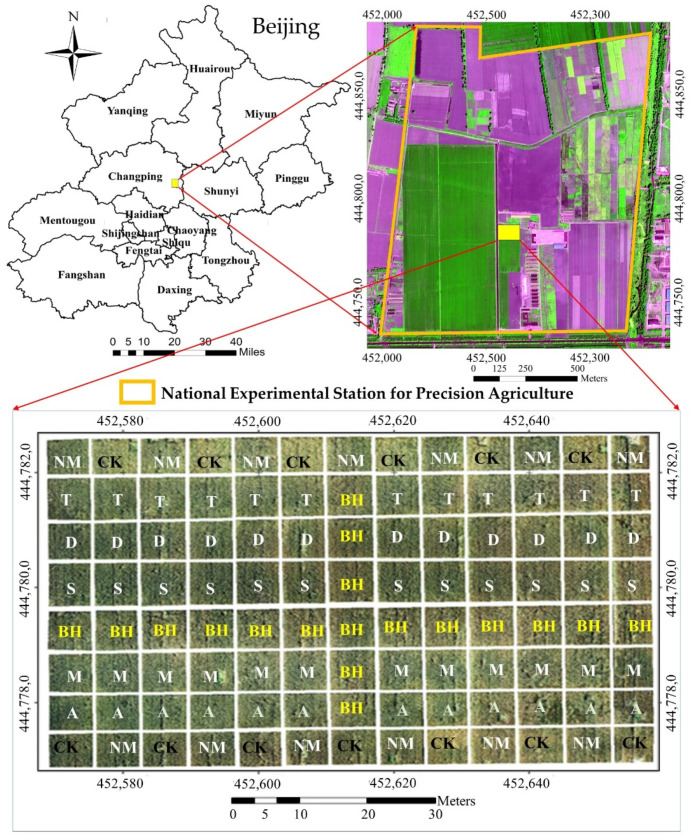
Study area and experimental design. NM treatment: A recommended conventional, uniform nitrogen fertilization treatment; CK treatment: non-nitrogen fertilizer treatment; BH treatment: excess nitrogen fertilizer treatment (N-rich strip treatment); S treatment: variable-rate nitrogen treatment with fertilizer rate based on the SPAD value ratio between this treatment and N-rich strip treatment; A treatment: variable-rate nitrogen treatment with fertilizer rate based on the ASD spectroradiometer vegetation index OSAVI value ratio between this treatment and N-rich strip treatment; T treatment: variable-rate nitrogen treatment with fertilizer rate based on the ASD spectroradiometer vegetation index REPLI value ratio between this treatment and N-rich strip treatment; D treatment: variable-rate nitrogen treatment with fertilizer rate based on the Dualex NBI value between this treatment and N-rich strip treatment; and M treatment: variable-rate nitrogen treatment with fertilizer rate based on the Multiplex NBI_R value ratio between this treatment and N-rich strip treatment.

**Figure 2 sensors-22-00549-f002:**
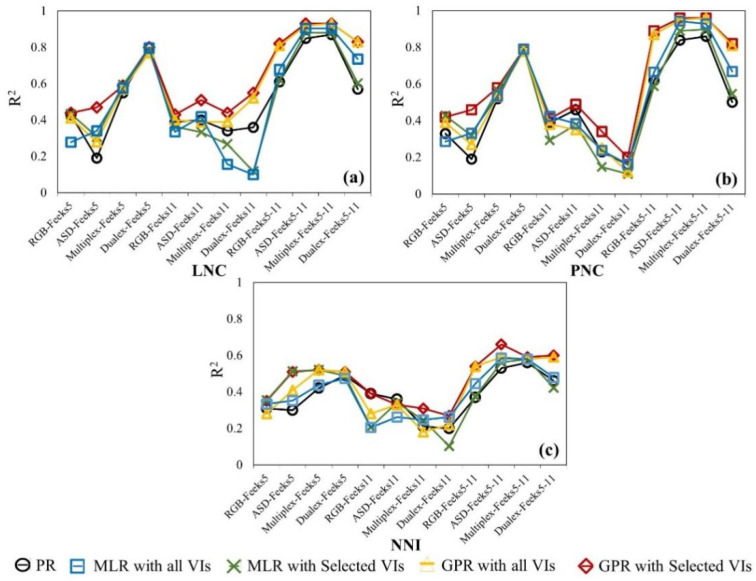
Cross-validation R^2^ for LNC (**a**), PNC (**b**), and NNI (**c**) estimation with different sensors through PR, MLR, and GPR methods.

**Figure 3 sensors-22-00549-f003:**
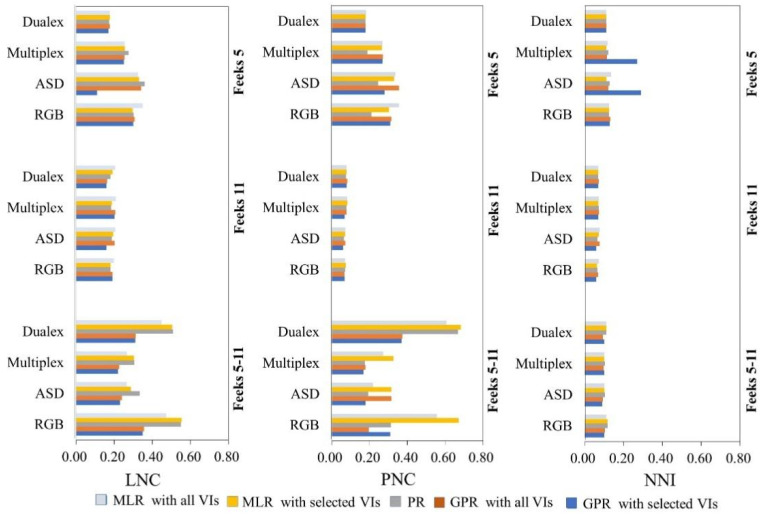
The MAE for LNC (**left**), PNC (**middle**), and NNI (**right**) estimation with different sensors through PR, MLR, and GPR methods.

**Figure 4 sensors-22-00549-f004:**
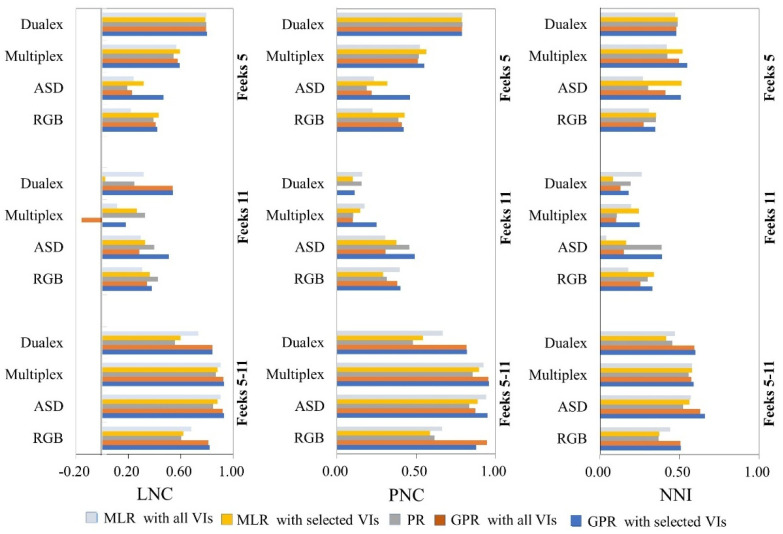
The NSE for LNC (**left**), PNC (**middle**), and NNI (**right**) estimation with different sensors through PR, MLR, and GPR methods.

**Figure 5 sensors-22-00549-f005:**
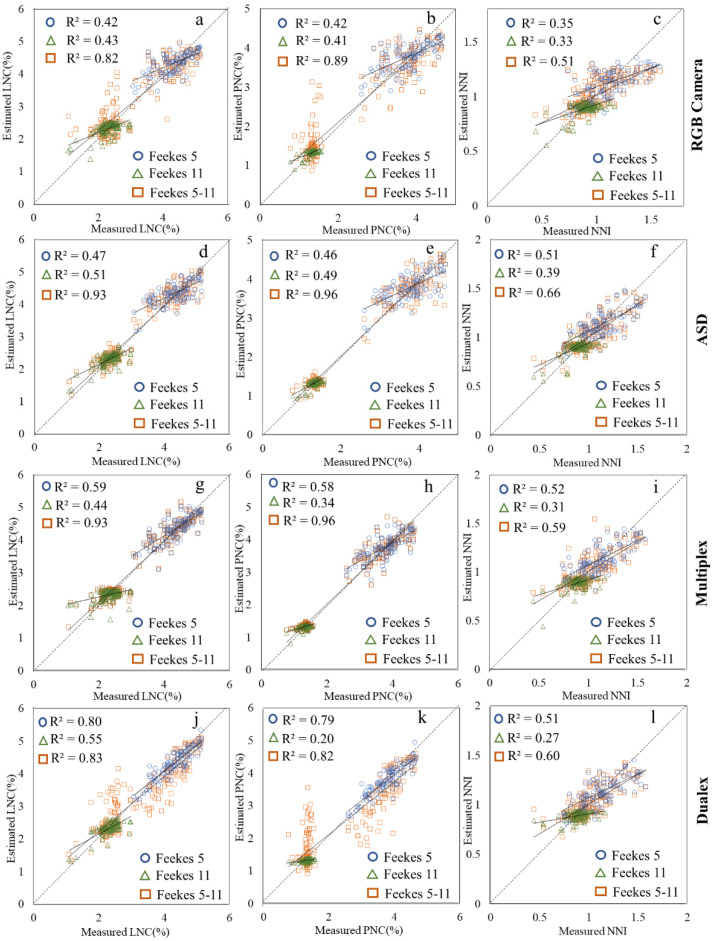
Measured vs. estimated LNC (left), PNC (middle), and NNI (right) values along the 1:1 line through the GPR-SBBR method based on best VIs for four different sensors. (**a**) LNC based on RGB sensor; (**b**) PNC based on RGB sensor; (**c**) NNI based on RGB sensor; (**d**) LNC based on ASD sensor; (**e**) PNC based on ASD sensor; (**f**) NNI based on ASD sensor; (**g**) LNC based on Multiplex sensor; (**h**) PNC based on Multiplex sensor; (**i**) NNI based on Multiplex sensor; (**j**) LNC based on Dualex sensor; (**k**) PNC based on Dualex sensor; and (**l**) NNI based on Dualex sensor.

**Table 1 sensors-22-00549-t001:** Fertilizer information for different treatments.

Treatment	Plot	Base Fertilizer		Topdressing Fertilizer		Fertilizer Treatment Rate Statistic			
	Number	Time	N kg/ha	Time	N kg/ha	Mean kg/ha	Min kg/ha	Max kg/ha	CV %
BH	18	Seed	72	Feekes 2, 4	51,102	225	225	225	0
NM	13	Seed	72	Feekes 6	78	150	150	150	0
CK	13	Seed	0	Feekes 6	0	0	0	0	0
A	12	Seed	72	Feekes 6	78	150	147	154.1	1.78
M	12	Seed	72	Feekes 6	78	150	138.6	162.2	5.15
D	12	Seed	72	Feekes 6	78	150	141.6	160	3.04
S	12	Seed	72	Feekes 6	78	150	144.7	154.8	1.92
T	12	Seed	72	Feekes 6	78	150	131	183.1	10.25

**Table 2 sensors-22-00549-t002:** Parameters on sensors used in this study.

Sensor Information	Polyphenol and Chlorophyll Meter	Polyphenol and Chlorophyll Meter	Field Spectrometer	UAV-Based Digital Camera
Sensor Type	Dualex	Multiplex	ASD	RGB Camera
Sensor name	Force-A Dualex Scientific	Force-A MULTIPLEX 3	ASD FieldSpec 4	Sony DSC–QX100
Target sample	Plant leaves	Plant canopy	Plant canopy	Plant canopy
Field of view	-	-	25°	64°
Image size	-	-	-	3000 × 4000
Working height	-	10 cm	1.3 m	50 m
Measurement area	5 mm in diameter	10 cm in diameter	50 cm in diameter	Full field
Spectral information	Excitation channels: UV (357 nm) and red (650 nm). Detection channels: red and far-red.	Excitation channels: UV (375 nm), blue (450 nm), green (510 nm), and red (630 nm). Detection channels: yellow, red, and far-red.	350–2500 nm	R,G,B
Original spectral resolution	-	-	3 nm @ 700 nm; 10 nm @ 1400 nm	-
Data spectral resolution			1 nm	
Image spatial resolution	-	-	-	2 cm

**Table 3 sensors-22-00549-t003:** Vegetation indices based on Multiplex 3 and ASD sensors used in this study.

Sensor	ID	Vegetation Index	Formula	Reference
Multiplex	1	SFR_G	FRF_G/RF_G	[17]
	2	SFR_R	FRF_R/RF_R	[17]
	3	BRR_FRF	BGF_UV/FRF_UV	[17]
	4	FER_RUV	FRF_R/FRF_UV	[17]
	5	FER_RG	FRF_R/FRF_G	[17]
	6	FLAV	Log(FER_RUV)	[17]
	7	ANTH	Log(FER_RG)	[17]
	8	NBI_G	FRF_UV/RF_G	[17]
	9	NBI_R	FRF_UV/RF_R	[17]
ASD	1	SR_(700,670)_	(R_700_)/(R_670_)	[40]
	2	SR_(418,450)_	(R_418_)/(R_450_)	[40]
	3	VOGa	(R_740_)/(R_720_)	[41]
	4	SR_(553,537)_	(R_553_)/(R_537_)	[42]
	5	NDCI	(R_762_ − R_527_)/(R_762_ + R_527_)	[41]
	6	NDRE	(R_790_ − R_720_)/(R_790_ + R_720_)	[43]
	7	TBI1	(R_434_)/(R_496_ + R_401_)	[44]
	8	mND_705_	(R_750_ − R_705_)/(R_750_ + R_705_ − 2R_445_)	[45]
	9	NDIopt	(R_503_ − R_483_)/(R_503_ + R_483_)	[46]
	10	TBI2	(R_924_ − R_703_)/(R_924_ − R_703_)	[47]
	11	NDVI_(670,780_	(R_780_ − R_670_)/(R_780_ + R_670_)	[48]
	12	RDVI	(R_800_ − R_670_)/(R_800_ + R_670_)^1/2^	[49]
	13	SR_(750,700)_	(R_750_)/(R_700_)	[50]
	14	WI	(R_900_)/(R_950_)	[51]
	15	NDWI	(R_860_ − R_1240_)/(R_860_ + R_1240_)	[52]
	16	NDII	(R_819_ − R_1600_)/(R_819_ + R_1600_)	[53]
	17	MCARI	[(R_700_ − R_670_) − 0.2(R_700_ − R_550_)](R_700_/R_670_)	[54]
	18	TCARI	3[(R_700_ − R_670_) − 0.2(R_700_ − R_550_)(R_700_/R_670_)]	[54]
	19	OSAVI	1.16(R_800_ − R_670_)/(R_800_ + R_670_ + 0.16)	[55]
	20	MSAVI	0.5[2R_800_ + 1 − ((2R_800_ + 1)^2^ − 8(R_800_ − R_670_))^1/2^]	[56]
	21	MCARI 1	1.2[2.5(R_800_ − R_670_) − 1.3(R_800_ − R_550_)]	[57]
	22	MCARI 2	1.5[2.5(R_800_ − R_670_) − 1.3(R_800_ − R_550_)]/[(2R_800_ + 1)^2^ − (6R_800_ − 5(R_670_)^1/2^) − 0.5]^1/2^	[57]
	23	PPR	(R_550_ − R_450_)/(R_550_ + R_450_)	[58]
	24	PVR	(R_550_ − R_650_)/(R_550_ + R_650_)	[59]
	25	PRI	(R_531_ − R_570_)/(R_531_ + R_570_)	[60]
	26	REP	700 + 40[(R_670_ + R_780_)/2) − R_700_)/(R_740_ − R_700_)]	[61]
	27	REV	Reflectance value at REP	[62]
	28	REFD	First deviation of red edge	[62]

**Table 4 sensors-22-00549-t004:** Vegetation indices based on digital RGB images.

ID	Vegetation Index	Full Name	Formula	Reference
1	R	DN values for red band	DN_R_/255	[64]
2	G	DN values for green band	DN_G_/255	[64]
3	B	DN values for blue band	DN_B_/255	[64]
4	r	Chromatic coordinates for red	R/(R + G + B)	[64]
5	g	Chromatic coordinates for green	G/(R + G + B)	[64]
6	b	Chromatic coordinates for blue	B/(R + G + B)	[64]
7	ExR	Excess red	1.4 × r − b	[65]
8	ExG	Excess green	2g − (r + b)	[65]
9	NDI	The normalized difference vegetation index	(b − g)/(b + g)	[65]
10	CVI1	Color vegetation index 1	(r − g)	[65]
11	CVI2	Color vegetation index 2	(g − b)	[65]
12	CVI3	Color vegetation index 3	(g − b)/(r − g)	[65]
13	GRVI	Green–red vegetation index	(g − r)/(g + r)	[65]
14	NPCI	Normalized pigment chlorophyll ratio index	(b − r)/(b + r)	[66]

**Table 5 sensors-22-00549-t005:** Statistics computed to compare the results of the different VIs used for the N parameter estimation.

Statistics	Formula	Character	Reference
Root mean square error (RMSE)	$\mathrm{RMSE}=\sqrt{\frac{\sum_{i}^{n}{\left(\hat{y_{i}}-y_{i}\right)}^{2}}{n}}$	from 0 to +∞, optimum 0	[71]
Mean absolute error (MAE)	$\mathrm{MAE}=\sum_{i=1}^{n}\left|\hat{y_{i}}-y_{i}\right|/n$	from 0 to +∞, optimum 0	[72]
Nash–Sutcliffe efficiency (NSE)	$\mathrm{NSE}=1-\sum_{i=1}^{n}{\left(y_{i-}\hat{y_{i}}\right)}^{2}/\sum_{i=1}^{n}{\left(y_{i}-\bar{y}\right)}^{2}$	from −∞ to 1, optimum 1	[73]

Notes:$\;\hat{y_{i}}$ is the predicted value of the ith observation, $y_{i}\;$ is the measured value of the ith observation, $\bar{y}$ is the average of the measured values, and n is the number of observations in the calibration set.

**Table 6 sensors-22-00549-t006:** Descriptive statistics of winter wheat LNC, PNC, and NNI across different growth stages.

Growth Stage	Parameter	Min	Max	Mean	Range	Std	CV (%)
Feekes 5	LNC (%)	3.07	5.16	4.33	2.09	0.49	11.30
	PNC (%)	2.60	4.67	3.77	2.07	0.51	13.42
	NNI	0.74	1.58	1.14	0.84	0.19	17.17
Feekes 11	LNC (%)	1.08	2.99	2.32	1.92	0.31	13.42
	PNC (%)	0.77	1.58	1.33	0.81	0.12	8.99
	NNI	0.44	1.17	0.89	0.73	0.11	11.97
Feekes 5–11	LNC (%)	1.08	5.16	3.33	4.08	1.08	32.34
	PNC (%)	0.77	4.67	2.55	3.90	1.26	49.46
	NNI	0.44	1.58	1.00	1.14	0.20	18.59

**Table 7 sensors-22-00549-t007:** Relationship between wheat N variables and VIs from different sensors at different growth stages (n = 104).

Sensor	Feekes Stage	LNC (%)				PNC (%)				NNI			
		Model	VI	R^2^	RMSE	Model	VI	R^2^	RMSE	Model	VI	R^2^	RMSE
RGB	5	Poly	NPCI	0.43 **	0.23	Poly	ExR	0.33 **	0.10	Poly	ExR	0.31 **	0.09
	11	Poly	NDI	0.39 **	0.38	Poly	NDI	0.39 **	0.26	Exp	NDI	0.36 **	0.16
	5–11	Poly	r	0.61 **	0.68	Pow	ExR	0.62 **	0.40	Log	ExR	0.37 **	0.16
ASD	5	Poly	NDIopt	0.19	0.44	Poly	NDIopt	0.19	0.30	Poly	SR_(553,537)_	0.30 **	0.16
	11	Poly	mND_705_	0.40 **	0.24	Poly	NDRE	0.46 **	0.09	Poly	NDRE	0.39 **	0.08
	5–11	Poly	NDIopt	0.85 **	0.43	Poly	NDIopt	0.84 **	0.26	Log	NDIopt	0.53 **	0.14
Multiplex	5	Poly	FLAV	0.55 **	0.33	Poly	FLAV	0.52 **	0.24	Poly	NBI_R	0.42 **	0.15
	11	Poly	SFR_R	0.34 **	0.25	Poly	SFR_R	0.23	0.11	Poly	SFR_R	0.21*	0.10
	5–11	Poly	BRR_FRF	0.87 **	0.39	Poly	BRR_FRF	0.86 **	0.24	Poly	NBI_G	0.56 **	0.13
Dualex	5	Log	NBI	0.80 **	0.22	Pow	NBI	0.79 **	0.23	Lin	NBI	0.49 **	0.14
	11	Poly	CHI	0.36 **	0.27	Log	CHI	0.16	0.11	Log	CHI	0.20 *	0.10
	5–11	Poly	NBI	0.57 **	0.72	Poly	NBI	0.50 **	0.92	Exp	NBI	0.46 **	0.15

Note: Lin: linear, Exp: exponential, Pow: power, Poly: second-order polynomial, Log: logarithmic; ** r (0.01, 70) = 0.302, indicates significance at the 0.01 probability level; * r (0.05, 70) = 0.232, indicates significance at the 0.05 probability level.

**Table 8 sensors-22-00549-t008:** Cross-validation results for N estimation through GPR and MLR based on UAV-mounted RGB camera VIs.

N	Feekes	VI	GPR-SBBR		MLR		VIs	GPR		MLR	
Variable	Stage		R^2^	RMSE	R^2^	RMSE		R^2^	RMSE	R^2^	RMSE
LNC	5	B, b	0.44	0.37	0.43	0.37	All 14 VIs	0.41	0.37	0.28	0.43
	11	NPCI	0.43	0.24	0.36	0.25	All 14 VIs	0.40	0.25	0.33	0.26
	5–11	B, g, CVI2	0.82	0.46	0.62	0.67	All 14 VIs	0.81	0.47	0.68	0.62
PNC	5	B, b	0.42	0.38	0.41	0.38	All 14 VIs	0.39	0.39	0.29	0.44
	11	G, NDI	0.41	0.09	0.29	0.10	All 14 VIs	0.38	0.09	0.40	0.09
	5–11	B, g, CVI2	0.89	0.43	0.59	0.82	All 14 VIs	0.87	0.45	0.67	0.74
NNI	5	B, b	0.35	0.16	0.35	0.16	All 14 VIs	0.28	0.16	0.33	0.16
	11	R, NDI, CVI2	0.33	0.09	0.34	0.09	All 14 VIs	0.33	0.09	0.26	0.10
	5–11	B, G, g, NDI, CVI2, CVI3	0.54	0.14	0.38	0.16	All 14 VIs	0.54	0.13	0.45	0.15

**Table 9 sensors-22-00549-t009:** Cross-validation results for N estimation through GPR and MLR methods based on ASD VIs.

N	Feekes	VI	GPR-SBBR		MLR		VIs	GPR		MLR	
Variable	Stage		R^2^	RMSE	R^2^	RMSE		R^2^	RMSE	R^2^	RMSE
LNC	5	TBI1, NDIopt, TBI2	0.47	0.36	0.31	0.41	All 28 VIs	0.28	0.42	0.34	0.42
	11	SR_(700,670)_, SR_(418,405)_, SR_(740,720)_	0.51	0.22	0.33	0.25	All 28 VIs	0.39	0.26	0.42	0.26
	5–11	SR_(418,405)_, TBI1, PPR, PRI, REFD	0.93	0.29	0.88	0.37	All 28 VIs	0.92	0.30	0.91	0.34
PNC	5	TBI1, NDIopt, TBI2	0.46	0.37	0.32	0.42	All 28 VIs	0.27	0.44	0.33	0.44
	11	SR_(418,405)_, NDIopt, RDVI, REFD	0.49	0.09	0.38	0.09	All 28 VIs	0.35	0.10	0.38	0.10
	5–11	TBI1, PPR, REV, REFD	0.96	0.27	0.89	0.42	All 28 VIs	0.95	0.28	0.94	0.31
NNI	5	MSAVI, PPR	0.51	0.14	0.51	0.14	All 28 VIs	0.41	0.15	0.35	0.17
	11	NDWI, MCARI, MSAVI, PVR	0.39	0.08	0.21	0.10	All 28 VIs	0.28	0.10	0.21	0.10
	5–11	SR_(418,405)_, NDIopt, MSAVI, MCARI1, PPR	0.66	0.12	0.56	0.13	All 28 VIs	0.59	0.12	0.59	0.13

**Table 10 sensors-22-00549-t010:** Cross-validation results for N estimation through GPR and MLR methods based on Multiplex VIs.

N	Feekes	VI	GPR-SBBR		MLR		VIs	GPR		MLR	
Variable	Stage		R^2^	RMSE	R^2^	RMSE		R^2^	RMSE	R^2^	RMSE
LNC	5	SFR_G, FLAV, NBI_R	0.59	0.31	0.59	0.31	All 9 VIs	0.57	0.31	0.58	0.32
	11	SFR_R	0.44	0.28	0.27	0.27	All 9 VIs	0.39	0.30	0.16	0.29
	5–11	SFR_R, BRR_FRF, NBI_R	0.93	0.29	0.88	0.38	All 9 VIs	0.93	0.30	0.90	0.34
PNC	5	SFR_R, FLAV, NBI_R	0.58	0.34	0.57	0.34	All 9 VIs	0.53	0.34	0.53	0.35
	11	SFR_R	0.34	0.10	0.15	0.11	All 9 VIs	0.25	0.11	0.24	0.11
	5–11	SFR_R, BRR_FRF, NBI_G	0.96	0.25	0.90	0.41	All 9 VIs	0.96	0.26	0.93	0.35
NNI	5	FLAV, FER_RG	0.52	0.34	0.52	0.14	All 9 VIs	0.52	0.14	0.44	0.15
	11	SFR_G, SFR_R	0.31	0.09	0.24	0.09	All 9 VIs	0.18	0.10	0.25	0.10
	5–11	SFR_G, SFR_R, NBI_G	0.59	0.13	0.58	0.13	All 9 VIs	0.58	0.13	0.58	0.13

**Table 11 sensors-22-00549-t011:** Cross-validation results through GPR and MLR methods at different wheat growth stages.

N	Feekes	VI	GPR-SBBR		MLR		VIs	GPR		MLR	
Variable	Stage		R^2^	RMSE	R^2^	RMSE		R^2^	RMSE	R^2^	RMSE
LNC	5	NBI	0.80	0.22	0.79	0.23	All 3 VIs	0.77	0.22	0.80	0.22
	11	Chl	0.55	0.21	0.12	0.31	All 3 VIs	0.52	0.21	0.10	0.36
	5–11	NBI, Chl	0.83	0.43	0.60	0.69	All 3 VIs	0.83	0.43	0.74	0.56
PNC	5	NBI	0.79	0.23	0.79	0.23	All 3 VIs	0.78	0.22	0.79	0.23
	11	Chl	0.20	0.11	0.11	0.11	All 3 VIs	0.12	0.12	0.16	0.11
	5–11	NBI, Chl	0.82	0.55	0.55	0.86	All 3 VIs	0.81	0.54	0.67	0.74
NNI	5	NBI	0.51	0.14	0.49	0.14	All 3 VIs	0.51	0.14	0.47	0.14
	11	Chl	0.27	0.10	0.10	0.10	All 3 VIs	0.22	0.10	0.26	0.09
	5–11	NBI, Chl	0.60	0.13	0.42	0.15	All 3 VIs	0.59	0.13	0.48	0.15

## Data Availability

The data are not publicly available due to the data also forms part of an ongoing study at this time.

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
