# Peer review of "Winter Wheat Nitrogen Estimation Based on Ground-Level and UAV-Mounted Sensors"

_sensors, 2022, doi:10.3390/s22020549_

Round 1
Reviewer 1 Report
The authors have used a GPR method to identify the best VIs sensitive to wheat LNC, PNC, and NNI for four different sensors. The observed wheat N parameters were well described by the GPR and traditional PR methods throughout the two growth stages. The results indicated that the optical fluorescence sensor provides more accurate estimates of winter wheat N status at early growth stage, while ASD sensors is good for N estimation at the late wheat growth. It is a well-designed experiment with potential impact for the greater research community. Recommend it for publication with minor revision.
Author Response
Comments and Suggestions for Authors
The authors have used a GPR method to identify the best VIs sensitive to wheat LNC, PNC, and NNI for four different sensors. The observed wheat N parameters were well described by the GPR and traditional PR methods throughout the two growth stages. The results indicated that the optical fluorescence sensor provides more accurate estimates of winter wheat N status at early growth stage, while ASD sensors is good for N estimation at the late wheat growth. It is a well-designed experiment with potential impact for the greater research community. Recommend it for publication with minor revision.
Response:We appreciate your comments and suggestions for our manuscript.
Reviewer 2 Report
The authors did a great job comparing the data collected from the four sensors detailed in the manuscript and on the analytical approaches to identify the best VIs related to N levels on winter wheat.
Below you will find some points that I believe could improve the quality of the manuscript and/or not clear for as I read through the manuscript.
- I understand that the manuscript is focused on the use of sensors and ways to analyze the data to identify the best one that can detect N differences on the winter wheat plant. I think that a discussion regarding the role of N on winter wheat yield and protein content could potentially enhance the quality of the manuscript. Differences in N status in the plant at a given point during the growing season do not necessarily translate into yield differences.
- Along those lines, on lines 362-365 the authors mentioned that the Dualex VIs could be used for N recommendations for winter wheat. I would disagree of such assessment since such recommendations are usually based on many years field data collected across different locations, including yield data. I believe the manuscript does not provide enough data to support that statement/conclusion.
- I would encourage the authors to explore ways to further summarize the data presented on the manuscript. There is a lot of numbers provided on some of the tables comparing different ways to analyze the data, different VIs, at different times during the growing season, and too many acronyms, which makes really difficult to understand the results being presented.
- There is not a lot of details regarding how the UAV data was processed to extract the data for the individual plots and treatments. When extracting the data from the individual plots, did the authors use some sort of segmentation to pull the data out from the vegetation fraction only or the data is a mix of vegetation+background? Could that have affected the results obtained on this study?
- In some parts of the manuscript when referring to either the performance of sensors or data analysis techniques (i.e. lines 305-06, 343-344), the authors make reference that some of those performed well even though R2 are quite low (range of 0.59-0.31). Statistics is not my strong suit, but I believe those would be considered good performances by a sensor. Maybe some clarification is warranted on that regard. Along the same lines, is the amount of data being analyzed enough to train machine learning algorithms such as PGR?
- On lines 324-325 the authors make a comment regarding the fact that they did a thorough assessment of leaf, canopy, and UAV level sensors in estimation of N status. I would disagree with that statement, especially when it comes to the UAV camera sensor, since there was only one sensor evaluated, it was flown at one altitude. There are many parameters (i.e. flight altitude and speed) that would had to be tested to support the statement made on the manuscript.
- Regarding to Figures 3 and 4, I think it would make easier for the reader to visualize the magnitude of the differences if all the graphs had the same scale on the X axis. Are the comparisons among values mentioned on the discussion based on visual observation or they are statistical differences? That was not clear to me, maybe I just missed that.
- Regarding Figure 4, the authors mentioned (lines 388-390) that the GPR methods have higher accuracy, especially when modelling with the ASD VIs across the wheat growth stages (Feekes 5-11). Looking at the LNC and PNC data, the RGB data looks like very similar to the ASD for the GPR methods. If one is making the same type of assessment over hundreds or thousands of plots, what would be author's recommendation? Maybe a bit of discussion on the practical application of those results would make it easier for the reader to understand some of those results.
- Regarding the English grammar, I think the manuscript is well written for the most part. There some minor typos (fields, line 141) and grammar structure (lines 346 and 350) that warrant a review of the document. At the same time, the authors can make sure that the appropriate units and unit notations are being used across the manuscript.
Author Response
Comments and Suggestions for Authors
The authors did a great job comparing the data collected from the four sensors detailed in the manuscript and on the analytical approaches to identify the best VIs related to N levels on winter wheat.
Response:We appreciate your comments and suggestions for our manuscript.
Below you will find some points that I believe could improve the quality of the manuscript and/or not clear for as I read through the manuscript.
- I understand that the manuscript is focused on the use of sensors and ways to analyze the data to identify the best one that can detect N differences on the winter wheat plant. I think that a discussion regarding the role of N on winter wheat yield and protein content could potentially enhance the quality of the manuscript. Differences in N status in the plant at a given point during the growing season do not necessarily translate into yield differences.
Response:Thank you for your comments and suggestions. Cereal crop nitrogen nutrition status in different growth stages is relevant to the crop yield and grain protein content. In this study, we try to reveal crop nitrogen nutrition status through different sensors, so we focus on evaluating the ability of these sensors on N detection. The effect of the crop nitrogen status on crop yield and grain protein content is important and complex, and we plan to analyze their relationship in our future work.
- Along those lines, on lines 362-365 the authors mentioned that the Dualex VIs could be used for N recommendations for winter wheat. I would disagree of such assessment since such recommendations are usually based on many years field data collected across different locations, including yield data. I believe the manuscript does not provide enough data to support that statement/conclusion.
Response:Thank you for your comments and suggestions. We modified this sentence to remove the recommendation part. The sentence ‘The Dualex VIs, both nitrogen balance index (NBI) and chlorophyll index (CHI), seem to be good indicators to evaluate the N conditions in wheat and to make recommendations for N management at the early growth stage.’ has been changed to ‘The Dualex VIs, both nitrogen balance index (NBI) and chlorophyll index (CHI), seem to be good indicators to evaluate the N conditions in wheat at the early growth stage.’
- I would encourage the authors to explore ways to further summarize the data presented on the manuscript. There is a lot of numbers provided on some of the tables comparing different ways to analyze the data, different VIs, at different times during the growing season, and too many acronyms, which makes really difficult to understand the results being presented.
- Response:Thank you for your comments and suggestions. We made some modifications and make the summary more clear and easier to understand for the reader.
- There is not a lot of details regarding how the UAV data was processed to extract the data for the individual plots and treatments. When extracting the data from the individual plots, did the authors use some sort of segmentation to pull the data out from the vegetation fraction only or the data is a mix of vegetation+background? Could that have affected the results obtained on this study?
Response:We added the detailed information about how to extract VIs from the UAV image for each plot. In this study, one area of interest (AOI), (2m × 2m) located in the middle region of each plot was selected from the RGB images. Then spectral data from the mix of winter wheat and soil background was extracted and the Vis were calculated to indicate the winter wheat growth status for each plot.
Soil or water background variations can cause changes in crop canopy spectral reflectance, which leads to uncertainty in estimating the canopy nitrogen (N) status. Some studies estimated the crop parameters by the pure vegetation pixels on plot level through removing the background pixels from the UAV images. In this study, we try to monitor the crop N status from leaf to canopy and UAV level by using different sensors. So we did not remove the soil background pixels from the UAV images in order to have a similar field of view for all canopy sensors (Multiplex3, ASD and UAV RGB camera).
- In some parts of the manuscript when referring to either the performance of sensors or data analysis techniques (i.e. lines 305-06, 343-344), the authors make reference that some of those performed well even though R2 are quite low (range of 0.59-0.31). Statistics is not my strong suit, but I believe those would be considered good performances by a sensor. Maybe some clarification is warranted on that regard. Along the same lines, is the amount of data being analyzed enough to train machine learning algorithms such as PGR?
Response: Thank you for your comments and suggestions. We indicate the significance level with ** (0.01 probability level) and * (0.05 probability level) for the parametric regression (PR) model in Table 7. The R2 listed in this table indicates the validation accuracy for the PR model with training data (randomly selected 70% samples for all 104). They show the sensors’ sensitivity to the crop N status at different growth stages.
In this study, the 10-fold cross-validation technique was used to determine the optimal number of latent factors based on the lowest root-mean-square error (RMSE) when estimating crop N parameters using GPR methods. All 104 data samples with different VIs collected by the four sensors were randomly divided into 10 parts with about the same size. The 10-fold cross-validation was done 10 times. In each stage, one fold gets to play the role of validation set whereas the other remaining parts (K-1) are the training set. Then the average standard error of all cross-validation error indicates the cross-validation estimate accuracy. So, I think the amount of data is enough to train machine learning algorithms in this study.
- On lines 324-325 the authors make a comment regarding the fact that they did a thorough assessment of leaf, canopy, and UAV level sensors in estimation of N status. I would disagree with that statement, especially when it comes to the UAV camera sensor, since there was only one sensor evaluated, it was flown at one altitude. There are many parameters (i.e. flight altitude and speed) that would had to be tested to support the statement made on the manuscript.
Response:Thank you for your comments and suggestions. We totally agree with you. Different types of sensors can be carried by the UAV to detect the crop growth and nutrition status. Crop information extracted from UAV images is affected not only by the sensor type but also by the UAV imaging conditions, such as UAV flight height and speed. In this study, we only used one sensor (a RGB camera) on the UAV to test its ability on crop N status estimation. We changed the sentence to ‘This allowed a preliminary assessment of the utility of the leaf, canopy and UAV-level sensors in estimation of N status indictors of winter wheat. The predictive performances of the four sensors were compared using different modeling methods (PR, MLR and GPR).’
It is necessary to make more thorough assessment for crop nitrogen status monitoring based on UAV sensors in different imaging conditions in our future study. Such a study will be very important and interesting for the estimation of N status.
- Regarding to Figures 3 and 4, I think it would make easier for the reader to visualize the magnitude of the differences if all the graphs had the same scale on the X axis. Are the comparisons among values mentioned on the discussion based on visual observation or they are statistical differences? That was not clear to me, maybe I just missed that.
- Response:Thank you for your comments and suggestions. The MAE and NSE for LNC, PNC and NNI were all calculated according to the formulas listed in Table 5. We modified the scale on the X axis of Figures 3 and 4 according to your suggestion, so the reader can easily compare the statistical differences of MAE and NSE values for all sensors at different wheat growth stages.
- Regarding Figure 4, the authors mentioned (lines 388-390) that the GPR methods have higher accuracy, especially when modelling with the ASD VIs across the wheat growth stages (Feekes 5-11). Looking at the LNC and PNC data, the RGB data looks like very similar to the ASD for the GPR methods. If one is making the same type of assessment over hundreds or thousands of plots, what would be author's recommendation? Maybe a bit of discussion on the practical application of those results would make it easier for the reader to understand some of those results.
- Response:Thank you for your comments and suggestions. We added some discussion for this point at Section 4.2.
- Regarding the English grammar, I think the manuscript is well written for the most part. There some minor typos (fields, line 141) and grammar structure (lines 346 and 350) that warrant a review of the document. At the same time, the authors can make sure that the appropriate units and unit notations are being used across the manuscript.
- Response:Thank you for your comments and suggestions. We have corrected the errors in lines 141, 346 and 350. We also checked the units and unit notations for the whole manuscript.
Reviewer 3 Report
The paper’s subject is relevant. Authors consider the problem of the practical application of UAV in agriculture. The problem and its decision are presented well. But I’d like to recommend increasing the introduction and some publications with the general analysis of the UAV applications in agriculture and other areas. For example, I’d like to recommend some of them:
Mukhamediev, R.I.; Symagulov, A.; et al Review of Some Applications of Unmanned Aerial Vehicles Technology in the Resource-Rich Country. Appl. Sci. 2021, 11, 10171.
Shakhatreh, H.; Sawalmeh, A.H.; Al-Fuqaha, A.; Dou, Z.; Almaita, E.; Khalil, I.; Othman, N.S.; Khreishah, A.; Guizani, M. Unmanned aerial vehicles (UAVs): A survey on civil applications and key research challenges. IEEE Access 2019, 7, 48572–48634.
Based on the reviews of UAVs applications the specific of their use in agriculture should be shortly presented.
One more non-significant comment. The sensor data collection is discussed in section 2.2. Could you discuss in more detail the influence of the different types of sensors on the presented result?
Author Response
Comments and Suggestions for Authors
The paper’s subject is relevant. Authors consider the problem of the practical application of UAV in agriculture. The problem and its decision are presented well. But I’d like to recommend increasing the introduction and some publications with the general analysis of the UAV applications in agriculture and other areas. For example, I’d like to recommend some of them:
Mukhamediev, R.I.; Symagulov, A.; et al Review of Some Applications of Unmanned Aerial Vehicles Technology in the Resource-Rich Country. Appl. Sci. 2021, 11, 10171.
Shakhatreh, H.; Sawalmeh, A.H.; Al-Fuqaha, A.; Dou, Z.; Almaita, E.; Khalil, I.; Othman, N.S.; Khreishah, A.; Guizani, M. Unmanned aerial vehicles (UAVs): A survey on civil applications and key research challenges. IEEE Access 2019, 7, 48572–48634.
Based on the reviews of UAVs applications the specific of their use in agriculture should be shortly presented.
Response: Thanks for your comments and suggestions. We added some introduction about the UAV application in agriculture and other fields in Sections 1 and 4.
One more non-significant comment. The sensor data collection is discussed in section 2.2. Could you discuss in more detail the influence of the different types of sensors on the presented result?
Response: Thanks for your comments and suggestions. We added the sub-title at Section 2.2 in order to clearly introduce how to collect data by different sensors. We also discussed their influence on crop N status estimation for different sensors at Section 4.
Reviewer 4 Report
The research article titled "Winter wheat nitrogen estimation based on ground-level and UAV-mounted sensors" has well written with clear objectives, proper methodology and strong results with discussion. The sensors data were properly analysed in achieving the results. The scope of this research may encourage many researchers to set their research objectives. The paper has well written, but some of the typographical errors are mentioned as below;
-Line 17: Check spacing “sensors were”
-Line 20: Check spacing “sensors. Wheat”
-Line 91: Correct “……. we proposed in this study to assess..” as “…… in this study, we proposed…..”
Line 98: Correct the spacing “PR,MLR”
Line 105: Correct as per journal format “2.1 Experimental Design”
Line 106: Revise “Our study was……” as “This study was…...”
Line 117-118: Provide high resolution of Figure 1
Line 170: Correct as per journal format “2.3 Plant Sampling data collection”
Line 172: Correct “0.12 m2”
Line 175: Correct “g/m2”
Line 184” Correct as “PNA (kg/ha) = (LB × LNC) + (SB × SNC) + (EB × ENC)”
Line 195: Correct as per journal format “2.4 Data Analysis methods”
Line 228: Correct “ith”
Line 233: revise as per journal format “3.1. Variation of Winter Wheat N Indicators”. Also, check other headings of the paper.
Line 249: Check spacing “stronger correlation” and “NNI.The”
Line 344: Check spacing “R2of 0.50”
Line 386: Revise “We can see from Figure 3 that the…” as “As shown in Figure 3, the…..”
Line 410: Add the reference at “Verrelst et al.”
Line 413: Check spacing “[35].In this study”
Author Response
Comments and Suggestions for Authors
The research article titled "Winter wheat nitrogen estimation based on ground-level and UAV-mounted sensors" has well written with clear objectives, proper methodology and strong results with discussion. The sensors data were properly analysed in achieving the results. The scope of this research may encourage many researchers to set their research objectives.
Response: Thanks for your comments and suggestions.
The paper has well written, but some of the typographical errors are mentioned as below;
-Line 17: Check spacing “sensors were”
Response: We checked and modified the error.
-Line 20: Check spacing “sensors. Wheat”
Response: We checked and modified the error.
-Line 91: Correct “……. we proposed in this study to assess..” as “…… in this study, we proposed…..”
Response: We corrected this sentence according the reviewer’s suggestion.
Line 98: Correct the spacing “PR,MLR”
Response: We checked and modified the error.
Line 105: Correct as per journal format “2.1 Experimental Design”
Response: We checked all the subtitle of the manuscript and modified them as the journal format.
Line 106: Revise “Our study was……” as “This study was…...”
Response: We corrected this sentence according the reviewer’s suggestion.
Line 117-118: Provide high resolution of Figure 1
Response: Yes, we provide high resolution figures of this manuscript for the Sensors editorial office.
Line 170: Correct as per journal format “2.3 Plant Sampling data collection”
Response: We checked all the subtitle of the manuscript and modified them as the journal format.
Line 172: Correct “0.12 m2”
Response: We checked and modified the error.
Line 175: Correct “g/m2”
Response: We checked and modified the error.
Line 184” Correct as “PNA (kg/ha) = (LB × LNC) + (SB × SNC) + (EB × ENC)”
Response: We made correction according to the reviewer’s suggestion.
Line 195: Correct as per journal format “2.4 Data Analysis methods”
Response: We checked all the subtitle of the manuscript and modified them as the journal format.
Line 228: Correct “ith”
Response: We made correction according to the reviewer’s suggestion.
Line 233: revise as per journal format “3.1. Variation of Winter Wheat N Indicators”. Also, check other headings of the paper.
Response: We checked all the subtitle of the manuscript and modified them as the journal format.
Line 249: Check spacing “stronger correlation” and “NNI.The”
Response: We checked and modified the error.
Line 344: Check spacing “R2of 0.50”
Response: We checked and modified the error.
Line 386: Revise “We can see from Figure 3 that the…” as “As shown in Figure 3, the…..”
Response: We made correction according to the reviewer’s suggestion.
Line 410: Add the reference at “Verrelst et al.”
Response: Yes, we add the reference of Verrelst et al. at the end of this sentence.
Line 413: Check spacing “[35].In this study”
Response: We checked and modified the error.
Round 2
Reviewer 2 Report
I appreciate the authors responses to my questions and comments raised during the first review of the manuscript. In addition, I appreciate the fact that most of the key comments/issues have been addressed on the second version of the manuscript. I believe the changes made to the current version of the document improved the overall quality of the manuscript.
I have only two small suggestions:
1) on Figure 4 the legend for the X axis for LNC seems to be two crowded because of both little space and the two decimals used on X axis values. Could the authors make graph wider or use less decimals on the X values so the values are not on top of each other?
2) Line 497 - it should read "UAV flight heights" instead of "UVA flight heights".
Author Response
Comments and Suggestions for Authors
I appreciate the authors responses to my questions and comments raised during the first review of the manuscript. In addition, I appreciate the fact that most of the key comments/issues have been addressed on the second version of the manuscript. I believe the changes made to the current version of the document improved the overall quality of the manuscript.
Response:We appreciate your comments and suggestions for our manuscript.
I have only two small suggestions:
- on Figure 4 the legend for the X axis for LNC seems to be two crowded because of both little space and the two decimals used on X axis values. Could the authors make graph wider or use less decimals on the X values so the values are not on top of each other?
Response: Thank you for your comments and suggestions. We modified the scale on the X axis of Figures 3 and 4 according to your suggestion, so the reader can easily compare the statistical differences of MAE and NSE values for all sensors at different wheat growth stages.
- Line 497 - it should read "UAV flight heights" instead of "UVA flight heights".
Response: Thank you for your comments and suggestions. We change the "UAV flight heights" to "UVA flight heights"
